

# Event-related potentials evoked by skin puncture reflect activation of Aβ fibers: comparison with intraepidermal and transcutaneous electrical stimulations

Yui Shiroshita[1], Hikari Kirimoto[2], Tatsunori Watanabe[2], Keisuke Yunoki[2] and Ikuko Sobue[1]

[1] Department of Nursing Science, Graduate School of Biomedical and Health Sciences, Hiroshima University, Hiroshima, Japan
[2] Department of Sensorimotor Neuroscience, Graduate School of Biomedical and Health Sciences, Hiroshima University, Hiroshima, Japan

Corresponding author
Hikari Kirimoto,
hkirimoto@hiroshima-u.ac.jp

## ABSTRACT

**Background**. Recently, event-related potentials (ERPs) evoked by skin puncture, commonly used for blood sampling, have received attention as a pain assessment tool in neonates. However, their latency appears to be far shorter than the latency of ERPs evoked by intraepidermal electrical stimulation (IES), which selectively activates nociceptive Aδ and C fibers. To clarify this important issue, we examined whether ERPs evoked by skin puncture appropriately reflect central nociceptive processing, as is the case with IES.

**Methods**. In Experiment 1, we recorded evoked potentials to the click sound produced by a lance device (click-only), lance stimulation with the click sound (click+lance), or lance stimulation with white noise (WN+lance) in eight healthy adults to investigate the effect of the click sound on the ERP evoked by skin puncture. In Experiment 2, we tested 18 heathy adults and recorded evoked potentials to shallow lance stimulation (SL) with a blade that did not reach the dermis (0.1 mm insertion depth); normal lance stimulation (CL) (1 mm depth); transcutaneous electrical stimulation (ES), which mainly activates Aβ fibers; and IES, which selectively activates Aδ fibers when low stimulation current intensities are applied. White noise was continuously presented during the experiments. The stimulations were applied to the hand dorsum. In the SL, the lance device did not touch the skin and the blade was inserted to a depth of 0.1 mm into the epidermis, where the free nerve endings of Aδ fibers are located, which minimized the tactile sensation caused by the device touching the skin and the activation of Aβ fibers by the blade reaching the dermis. In the CL, as in clinical use, the lance device touched the skin and the blade reached a depth of 1 mm from the skin surface, *i.e.*, the depth of the dermis at which the Aβ fibers are located.

**Results**. The ERP N2 latencies for click-only (122 ± 2.9 ms) and click+lance (121 ± 6.5 ms) were significantly shorter than that for WN+lance (154 ± 7.1 ms). The ERP P2 latency for click-only (191 ± 11.3 ms) was significantly shorter than those for click+lance (249 ± 18.6 ms) and WN+lance (253 ± 11.2 ms). This suggests that the click sound shortens the N2 latency of the ERP evoked by skin puncture. The ERP N2 latencies for SL, CL, ES, and IES were 146 ± 8.3, 149 ± 9.9, 148 ± 13.1, and 197 ± 21.2 ms, respectively. The ERP P2 latencies were 250 ± 18.2, 251 ± 14.1,

237 ± 26.3, and 294 ± 30.0 ms, respectively. The ERP latency for SL was significantly shorter than that for IES and was similar to that for ES. This suggests that the penetration force generated by the blade of the lance device activates the Aβ fibers, consequently shortening the ERP latency.

**Conclusions**. Lance ERP may reflect the activation of Aβ fibers rather than Aδ fibers. A pain index that correctly and reliably reflects nociceptive processing must be developed to improve pain assessment and management in neonates.

## INTRODUCTION

Blood sampling is an essential medical procedure, yet the pain sensation associated with skin puncture is often a problem. In particular, neonates in neonatal intensive care units frequently experience blood sampling by skin puncture (*Cruz, Fernandes & Oliveira, 2016*), and this frequent exposure to unanticipated external nociceptive stimuli has been suggested to adversely affect the neuronal developmental process (*Ranger & Grunau, 2014*; *Walker, 2019*). Repeated painful procedures in neonates reduce the volume of white and gray matter (*Brummelte et al., 2012*) and lead to behavioral abnormalities (*Grunau et al., 2009*; *Vinall et al., 2014*); these adverse effects may last until adolescence (*Nosarti et al., 2002*; *Nosarti et al., 2008*; *Anderson & Doyle, 2003*; *Grunau, Whitfield & Fay, 2004*; *Schmidt et al., 2010*; *Loe et al., 2011*; *Lax et al., 2013*; *Walker et al., 2018*). Many researchers have attempted to relieve the pain of skin puncture in the heel in neonates (*Pillai Riddell et al., 2015*; *Stevens et al., 2017*) using interventions such as pacifiers, holding, music, and a combination of these (*Gao et al., 2018*; *Peng et al., 2018*; *Perroteau et al., 2018*; *Uematsu & Sobue, 2019*; *Campbell-Yeo, 2019*; *Davari et al., 2019*). However, some have suggested that the Premature Infant Pain Profile (*Stevens et al., 1996*; *Gibbins et al., 2014*) used to evaluate neonatal pain in these studies has issues in terms of pain detection sensitivity (*Hartley et al., 2015*). Indeed, an index that can be applied to objectively and quantitatively evaluate pain in neonates has yet to be established.

Recently, event-related potentials (ERPs) evoked by nociceptive stimuli have received increasing attention as a pain index in neonates (*Slater et al., 2010a*; *Slater et al., 2010b*; *Moultrie, Slater & Hartley, 2017*; *Shiroshita et al., 2020*). However, this index leaves doubts about the research results. In neonates, the ERP evoked by skin puncture in the heel (heel lance) consists of N2P2 and N3P3 waves, and the N3P3 is considered an ERP specific to heel lance (*Shiroshita et al., 2021*). The N3P3 latency has been reported as 420 ms (*Slater et al., 2010b*), 383 ms (*Verriotis et al., 2016*), and 403 ms (*Fabrizi et al., 2016*) for N3, and as 560 ms (*Slater et al., 2010b*), 554 ms (*Verriotis et al., 2016*), and 538 ms (*Fabrizi et al., 2016*) for P3. However, studies have yet to be conducted that investigate the response to intraepidermal electrical stimulation (IES) or laser stimulation in neonates. Laser stimulation is widely used to generate thermal stimuli and selectively activate heat-sensitive

nociceptors (*Bromm, Jahnke & Treede, 1984*). IES, as an alternative to laser stimulation, avoids limitations such as skin overheating and lesions due to repeated laser stimulation; moreover, it can selectively activate nociceptive Aδ and C fibers in the epidermal layer of the skin when low stimulation current intensities are applied (*Inui et al., 2002*; *Mouraux, Iannetti & Plaghki, 2010*). However, the exact latency evoked by the activation of Aδ and C fibers in neonates remains unknown, and whether the ERP evoked by heel lance in neonates reflects activation of Aδ or C fiber is yet to be determined (*Shiroshita et al., 2021*).

In previous studies of skin puncture in adults in which the same lance device was used as that in a neonatal heel lance, the latency of the ERP in response to lance stimulation at the hand finger (N2: 130 ± 40 ms, P2: 258 ± 61 ms (*Fabrizi et al., 2013*); N2: 102 ms, P2: 249.5 ms (*Fabrizi et al., 2016*)) was shorter than that of IES at the hand dorsum (N2: 199–232 ms, P2: 302–377 ms (*Inui et al., 2002*; *Mouraux, Iannetti & Plaghki, 2010*; *Otsuru et al., 2010*; *Kodaira, Inui & Kakigi, 2014*; *Mouraux, Marot & Legrain, 2014*; *Omori et al., 2017*; *Kirimoto et al., 2018*)) and laser stimulation at the hand dorsum (N2: 185–274 ms, P2: 277–399 ms (*Mouraux, Iannetti & Plaghki, 2010*; *Otsuru et al., 2010*; *Lefaucheur et al., 2012*)). The latency of the lance ERP appears to be rather close to the latency of ERPs generated in response to non-nociceptive nerve stimulation, which mainly activates the Aβ fibers at the hand dorsum (N2: 134–147 ms, P2: 235–293 ms (*Inui et al., 2002*; *Mouraux, Iannetti & Plaghki, 2010*; *Otsuru et al., 2010*)). It has been reported that thermal, electrical, and mechanical stimuli have different transduction times due to the characteristics of their receptor activation. Aδ fibers activated by laser stimulation are delayed relative to the direct electrical stimulation of IES because skin receptors are excited *via* temperature conduction (*Bromm, Jahnke & Treede, 1984*; *Inui et al., 2002*). Mechanical stimulation makes it difficult to avoid activation of Aβ fibers with low thresholds and fast conduction velocities (*Baumgärtner, Greffrath & Treede, 2012*). Indeed, the blade of the lance stimulation device reaches a depth of 1 mm from the skin surface (the dermis layer); thus, it penetrates the 0.2 mm thick epidermal layer where the terminal ends of Aδ and C fibers are located (*Novotny & Gommert-Novotny, 1988*; *Inui et al., 2002*). In addition, during the skin puncture procedure, the lance stimulation device is pressed against the puncture site and a button on the device is pressed to push out the blade. We speculate that these features and actions can cause a tactile or vibration sensation, resulting in the activation of the Aβ fibers, which consequently shortens the latency of the lance ERP. A previous finding that latency of ERP evoked by transcutaneous electrical stimulation (ES) activating both Aδ and Aβ fibers is shorter than that evoked by a laser stimulation appears to support this speculation (*Hird et al., 2018*). In addition, lance stimulation produces a click sound when a button on the device is pressed to push out the blade, which potentially affects the lance ERP. Specifically, simple short auditory stimuli have been reported to produce ERP components at around 100 ms and 180 ms post-stimuli (*Wolpaw & Penry, 1975*; *Näätänen & Picton, 1987*; *Martin, Tremblay & Korczak, 2008*; *Touge et al., 2008*; *Sakamoto, Nakata & Kakigi, 2009*). Hence, in addition to the activation of the Aβ fibers, we speculate that the click sound produced by the lance device influences the ERP evoked by skin puncture. It is possible that the lance stimulation used commonly in clinical practice does not evoke ERPs reflecting nociceptive processing (*i.e.,* Aδ and C fibers).
To reduce the Aβ fiber activation, we created a condition in which tactile pressure and vibration were excluded as much as possible by making a base on which to fix the lance stimulation device and by having a shallower blade insertion depth than the depth used in clinical practice. Although our interest is in neonates, it is impossible to evaluate their sensory thresholds and determine an appropriate ES intensity since they cannot express their feelings verbally. Therefore, healthy adults were used to investigate whether the lance ERP would be appropriate as a parameter for the objective evaluation of nociceptive processing. We conducted two experiments. In Experiment 1, we investigated the effect of the click sound on the ERP evoked by skin puncture. In Experiment 2, we investigated whether Aβ fiber activation is associated with the lance ERP.

## MATERIALS & METHODS

### Subjects

Eighteen healthy volunteers (13 males and five females; 19–34 years old) participated in this study (eight volunteers in Experiment 1 and 18 in Experiment 2). Their height was 153.0–180.0 cm (169.3 ± 7.1 cm). They were not undergoing any medical treatments and did not take analgesic drugs within the 48 h prior to the experiment. The study was approved by the Ethics Committee of Hiroshima University (approval number: E-2044) and written consent was obtained from all subjects.

### Experimental procedure

#### Experiment 1

Each subject comfortably sat in a recliner with a head pillow and leg rests and placed his/her right arm on the armrest in a neutral position. We recorded evoked potentials (EPs) in response to the click sound produced by a lance device (click-only), shallow lancing (SL) with the click sound (click+SL), and SL with white noise (WN+SL). Each of these was performed in a random order during one day. SL was applied to the dorsum of the right hand between the first and second metacarpal bones, and the stimulations were performed within 2 × 2 cm. For click-only, the click sound was produced near the dorsum of the right hand. To avoid habituation, the three stimulation types were performed at intervals of a few minutes.

*Three types of stimulation*

*Click-only.* We used a lancet (BD Microtainer Quikheel™ Lancet 368102, Japan Becton, Dickinson, Japan) that is clinically used for the heel lance of neonates. The lance device produces a click sound when a button on the device is pressed. The click sound was produced four times near the dorsum of the right hand without skin puncture.

*Click+SL.* The lancet housed a 2.5 mm spring-loaded blade, which is released by pressing a button on the top of the device. After creating the incision, the blade automatically and immediately retracts back inside the device. The original penetration depth was 1 mm. To minimize the tactile sensation caused by the device touching the skin and the activation of Aβ fibers by the blade reaching the dermis (*Munger & Halata, 1983*; *Mouraux, Iannetti &*

(A)    (B)

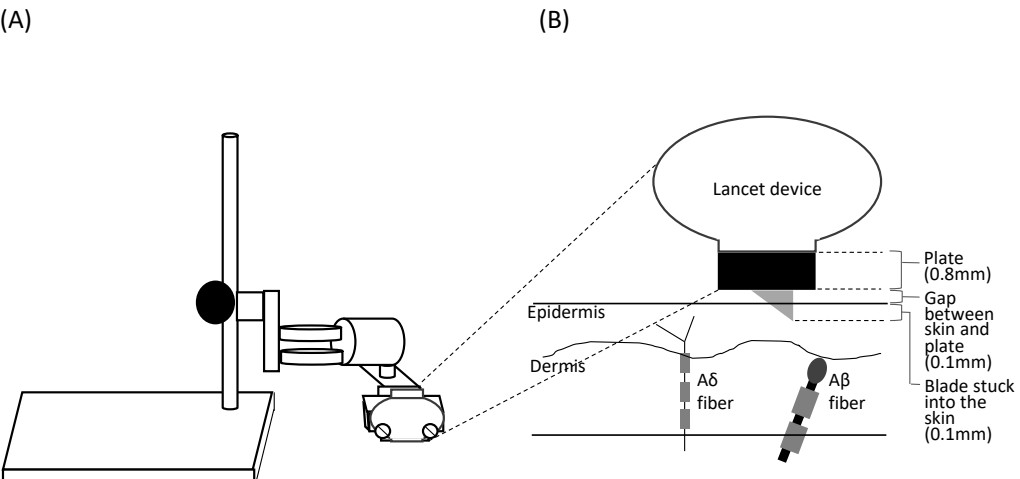

**Figure 1 Schema of the stand used for shallow lance stimulation (A) and insertion of a blade into the skin layers (B).** The lancet device was fixed using a fixing stand for shallow lance stimulation. A space for a metal plate (0.8 mm) and a small gap (about 0.1 mm) was made between the device and the skin surface. A blade was inserted into the epidermis. Free endings of Aδ fibers are present in the epidermis, and Aβ fiber receptors are present in the dermis (*Munger & Halata, 1983*; *Novotny & Gommert-Novotny, 1988*; *Inui et al., 2002*).

*Plaghki, 2010*), we placed a 0.8 mm plate between the device and the skin; we also fixed the device to the plate using a fixing stand (NISSIN SEIKI CO., LTD., Japan), which created a space of approximately 0.1 mm between the plate and the skin surface (Fig. 1). Thus, the blade was inserted to a depth of 0.1 mm into the epidermis, where the free nerve endings of Aδ fibers are located (*Novotny & Gommert-Novotny, 1988*; *Inui et al., 2002*). We performed the click+SL procedure four times. The stimulus location was changed for each stimulation to avoid habituation.

*WN+SL.* The white noise was presented over headphones. We confirmed that the subjects did not hear the click sound from the lance device. We performed the stimulation at least four times and added the number of times when the peak-to-peak amplitude of the EP was below 20 μV (4–5 times). The stimulus location was changed for each stimulation to avoid habituation.

### Experiment 2

Each subject comfortably sat in a recliner with a head pillow and leg rests and placed his/her right arm on the armrest in a neutral position. The subjects received four types of stimulation, *i.e.,* SL, clinical lance stimulation (CL), ES, and IES, randomly in one day. White noise was presented over headphones for all the stimulation conditions because we found an effect of the click sound on the ERP evoked by skin puncture (as described below). We confirmed that the subjects did not hear the click sound during the stimulation. The stimulation was applied to the dorsum of the right hand between the first and second metacarpal bones, and all stimulations were performed within 2 × 2 cm. To avoid habituation, the four stimulation types were provided with intervals of a few minutes

between each stimulus. We recorded EPs elicited by the stimulation. To equalize attention across different stimulations, the subject was asked to report using a visual analog scale (VAS), with zero meaning "no pain" and 100 meaning "worst pain", for each stimulation.

*Four types of stimulation*

*SL.* The stimulation method was same as WN+SL in Experiment 1. We performed SL at least four times and added the number of times when peak-to-peak amplitude of the EP was below 20 µV (4–5 times).

*CL.* In CL, as in clinical use, the surface of the lancet touched the skin and the blade reached a depth of 1 mm from the skin surface, *i.e.,* the depth of the dermis where Aβ fibers are located. At least four times were performed (4–6 times) and the stimulus location was within 1 cm from the SL location.

*ES.* For tactile stimulation, we applied ES using a bipolar felt-tip electrode (Digitimer DS7; Digitimer, UK). The stimulus was a 1-ms train of triple square wave pulses delivered with an interstimulus interval (ISI) of 2 ms. The stimulus intensity was 1.2–1.5 times the sensory threshold ($0.8 \pm 0.3$ mA). The sensory threshold was determined as follows: we started with an intensity of 0.01 mA and increased it by 0.01 mA until the subject felt a tactile sensation; the intensity was then reduced by 0.01 mA until the subject's tactile sensation disappeared; we determined the stimulus intensity at which clear middle-to-late EPs were obtained without a pricking sensation (*Inui et al., 2002*) to be 1.2–1.5 times the sensory threshold. We performed 15 ESs with an interval of 7–9 s. None of the subjects reported pinprick pain in response to the ES.

*IES.* For nociceptive stimulation, we used IES, which selectively activates Aδ fibers when low stimulation current intensities are applied (*Inui et al., 2002*; *Mouraux, Iannetti & Plaghki, 2010*). The stimulus was delivered using a stimulator (PNS-7000; Nihon Kohden) and a stainless steel concentric bipolar needle electrode (NM-980W; Nihon Kohden). The stimulus was a train of triple triangular wave pulses (a rise and fall time of 0.5 ms) with an ISI of 5 ms. The stimulus intensity was 1.5–2.25 times the sensory threshold ($0.07 \pm 0.02$ mA). The sensory threshold was determined as follows: we started with an intensity of 0.01 mA and increased it by 0.01 mA until the subject felt a pricking sensation; the intensity was then reduced by 0.01 mA until the subject's pricking sensation disappeared; we determined the stimulus intensity at which subjects felt pricking sensations and obtained clear middle-to-late EPs (*Inui et al., 2002*) at 1.5–2.25 times the sensory threshold. We performed 15 IESs with an interval of 10–15 s.

*Sample size calculation.* The sample size was calculated using the following formula:

$$n = \frac{\lambda^2 C^2}{e^2} = 17.59 \tag{1}$$

where λ is 1.96 (95% confidence interval), $C$ is the coefficient of variance (0.107) obtained from our previous study (*Kirimoto et al., 2018*), and $e$ is the acceptable error rate of 0.05. Based on this calculation, the sample size was set as 18 in the present study.

## Recording EPs

EPs were recorded using an amplifier (FA-DL-160, 4 Assist, Japan) with Cz (International 10–20 system) as the active electrode because the maximum response has previously been recorded from the Cz derivation in both electrical and lance stimulations (*Kakigi, Shibasaki & Ikeda, 1989*; *Slater et al., 2010b*; *Otsuru et al., 2010*). The earlobe (A2) was used as a reference. For the ground electrode, a disposable gel electrode was placed on the right forearm (GE Health Care Japan, Tokyo, Japan). The recording was made at a sampling rate of 4,000 Hz with a bandpass filter of 0.1–50.0 Hz. In Experiment 1, average waveforms were created from at least four artifact-free EPs (4–5 times) for the click-only, click+SL, and WN+SL conditions. In Experiment 2, average waveforms were created from 12 artifact-free EPs for ES and IES, and from at least four artifact-free EPs (4–6 times) for SL and CL. The EPs for the click-only, click+SL, WN+SL, SL, and CL were time-locked to the time at which an accelerometer (FA-DL-111; 4 Assist, Japan) attached to the experimenter's finger detected the start of lancet button pressing (movement of finger); when the button was pressed, the blade was released from the device to incise the skin. The time lag from when the lancet button started to be pressed until the blade reached the skin was about 2 ms or less as measured by a high-speed camera (1,200 frames/s), which was considered negligible here.

## Data and statistical analysis

ES and IES evoked negative–positive waveforms (N2–P2) with different latencies. The peak latencies of N2 and P2 for the ES were measured between 120–180 ms and 180–280 ms, respectively, and these values for the IES were measure between 170–250 ms and 250–350 ms, respectively (*Inui et al., 2002*; *Mouraux, Iannetti & Plaghki, 2010*; *Otsuru et al., 2010*; *Kodaira, Inui & Kakigi, 2014*; *Mouraux, Marot & Legrain, 2014*; *Omori et al., 2017*; *Kirimoto et al., 2018*). Click-only, click+SL, WN+SL, SL, and CL also evoked N2–P2. For the click-only, click+SL, WN+SL, SL, and CL conditions, we created the grand average of individual ERPs across all subjects and then selected the N2 and P2 peaks in the individual ERP waveforms that were located closest to those in the grand average (*Fabrizi et al., 2013*).

In Experiment 1, we compared the N2 and P2 latencies and N2–P2 amplitudes among the three stimulation conditions (*i.e.,* click-only, click+SL, and WN+SL) using a one-way repeated-measures ANOVA. In Experiment 2, we compared the N2 and P2 latencies and N2–P2 amplitudes among the four stimulation conditions (*i.e.,* SL, CL, ES, and IES) using a one-way repeated-measures ANOVA. A Bonferroni correction was used for the *post hoc* analysis. The VAS score was compared among the four stimulation conditions (*i.e.,* SL, CL, ES, and IES) initially with a Friedman test and then with a Wilcoxon signed-rank test for *post hoc* analysis. The correlations between the VAS score and N2 or P2 latencies and between the VAS score and N2–P2 amplitudes were calculated using Spearman's rank correlation coefficients. SAS9.4 was used to perform all statistical analyses.
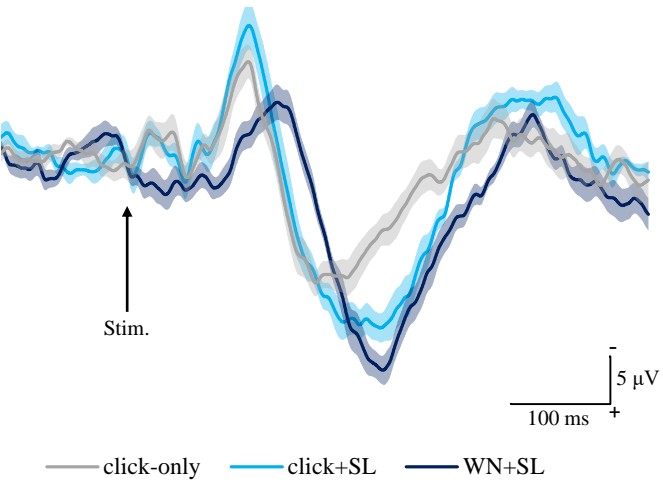

Stim.

5 μV

100 ms

———— click-only    ———— click+SL    ———— WN+SL

**Figure 2 Grand averaged ERPs recorded from Cz evoked by three stimulation condition (click-only, click+SL, and WN+SL) ($n = 8$).** click-only: click sound produced by a lance device; click+SL: shallow lance stimulation with click sound; WN+SL: shallow lance stimulation with white noise. Note the difference in ERP latency between the three types of stimulation. The shadow indicates the 95% confidence interval.

**Table 1 ERPs latencies and amplitudes for three stimulation conditions ($n = 8$).**

|  | Latency (ms) | | Amplitude (μV) | |
| --- | --- | --- | --- | --- |
|  | N2 | P2 | N2 | P2 |
| click-only | $122 \pm 2.9$ | $191 \pm 11.3$ | $-10.3 \pm 5.7$ | $14.8 \pm 5.0$ |
| click+SL | $121 \pm 6.5$ | $249 \pm 18.6$ | $-14.0 \pm 7.1$ | $21.4 \pm 4.9$ |
| WN+SL | $154 \pm 7.1$ | $253 \pm 11.2$ | $-6.8 \pm 5.2$ | $24.7 \pm 4.5$ |

Notes.
ERP, event-related potential; click-only, click sound produced by a lance device; click+SL, shallow lance stimulation with click sound; WN+SL, shallow lance stimulation with white noise.

## RESULTS

### Experiment 1

Figure 2 shows the grand averaged waveforms of potentials evoked by click-only, click+SL, and WN+SL (N2–P2). The mean and SD of the peak latencies for N2 and P2 are shown in Table 1.

The mean and individual latencies of N2 and P2 for each stimulation condition (click-only, click+SL, and WN+SL) are summarized in Fig. 3. A one-way repeated-measures ANOVA showed that the latencies of both N2 and P2 were significantly different among the three stimulation conditions (N2: $F_{(2,14)} = 120.3$, $P < 0.0001$, $\eta^2_p = 0.9450$; P2: $F_{(2,14)} = 50.89$, $P < 0.0001$, $\eta^2_p = 0.8791$). *Post hoc* analyses showed that N2 latencies for the click-only ($122 \pm 2.9$ ms) and click+SL ($121 \pm 6.5$ ms) were significantly shorter than that for the WN+SL ($154 \pm 7.1$ ms: $P < 0.0001$ for click-only; $P < 0.0001$ for click+SL). There was no significant difference in N2 latency between the click-only and the click+SL. The P2 latency for the click-only ($191 \pm 11.3$ ms) was significantly shorter than that

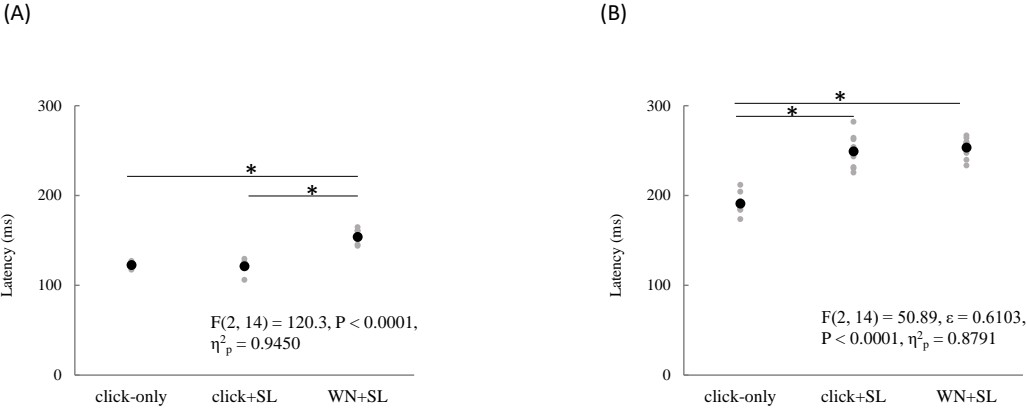

**Figure 3 N2 (A) and P2 latencies (B) for three stimulation condition (click-only, click+SL, and WN+SL).** Black and gray circles indicate the mean and the individual data, respectively. click-only: click sound produced by a lance device; click+SL: shallow lance stimulation with click sound; WN+SL: shallow lance stimulation with white noise. Asterisks indicate significant *post hoc* differences.

for the click+SL ($249 \pm 18.6$ ms: $P < 0.0001$) and that for the WN+SL ($253 \pm 11.2$ ms: $P < 0.0001$). There was no significant difference in P2 latency between the click+SL and the WN+SL.

A one-way repeated-measures ANOVA showed that the P2 amplitude was also significantly different among the three stimulation conditions ($F_{(2,14)} = 6.32$, $P = 0.0111$, $\eta^2_p = 0.4744$), but there was no significant difference in N2 amplitude among the three stimulation conditions (Table 1). *Post hoc* analyses showed that the P2 amplitude for the click-only condition ($14.8 \pm 5.0$ μV) was significantly shorter than those for the click+SL ($21.4 \pm 4.9$ μV: $P = 0.0422$) and WN+SL ($24.7 \pm 4.5$ μV: $P = 0.0257$). There was no significant difference in P2 amplitude between the click+SL and the WN+SL.

## Experiment 2

Figure 4 shows the grand averaged waveforms of potentials evoked by SL, CL, ES, and IES (N2–P2). The mean and SD of the peak latencies for N2 and P2 are shown in Table 2.

The mean and individual latencies of N2 and P2 for each stimulation condition (SL, CL, ES, and IES) are summarized in Fig. 5. A one-way repeated-measures ANOVA showed that the latencies of both N2 and P2 were significantly different among the four stimulation conditions (N2: $F_{(3,51)} = 65.51$, $P < 0.0001$, $\eta^2_p = 0.7940$; P2: $F_{(3,51)} = 40.54$, $P < 0.0001$, $\eta^2_p = 0.7045$). *Post hoc* analyses showed that N2 latencies for the SL ($146 \pm 8.3$ ms), CL ($149 \pm 9.9$ ms), and ES ($148 \pm 13.1$ ms) were significantly shorter than that for the IES ($197 \pm 21.2$ ms: $P < 0.0001$ for SL; $P < 0.0001$ for CL; $P < 0.0001$ for ES). The P2 latencies for the SL ($250 \pm 18.2$ ms), CL ($251 \pm 14.1$ ms), and ES ($237 \pm 26.3$ ms) were significantly shorter than that for the IES ($294 \pm 30.0$ ms: $P < 0.0001$ for SL; $P < 0.0001$ for CL; $P < 0.0001$ for ES). There was no significant difference in the N2 and P2 latencies between SL, CL, and ES.

A one-way repeated-measures ANOVA showed that the N2–P2 amplitudes were also significantly different among the four stimulation conditions ($F_{(3,51)} = 11.70$, $P < 0.0001$,

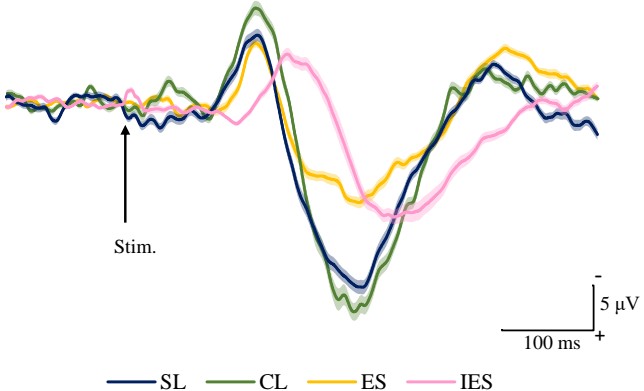

Stim.

SL — CL — ES — IES

**Figure 4** **Grand averaged ERPs recorded from Cz evoked by four stimulation condition (SL, CL, ES, and IES) ($n = 18$).** SL, shallow lance stimulation with white noise; CL, clinical lance stimulation with white noise; ES, transcutaneous electrical stimulation with white noise; IES, intraepidermal electrical stimulation with white noise. Note the difference in ERP latency between the four types of stimulation. The shadow indicates the 95% confidence interval.

**Table 2** **ERPs latencies, amplitudes and VAS scores for four stimulation conditions ($n = 18$).**

| | Latency (ms) | | Amplitude (µV) | VAS (points) |
|---|---|---|---|---|
| | N2 | P2 | N2-P2 | |
| SL | 146 ± 8.3 | 250 ± 18.2 | 33.0 ± 8.7 | 6.4 ± 4.7 |
| CL | 149 ± 9.9 | 251 ± 14.1 | 38.1 ± 11.5 | 17.9 ± 10.6 |
| ES | 148 ± 13.1 | 237 ± 26.3 | 22.6 ± 4.3 | 3.8 ± 3.7 |
| IES | 197 ± 21.2 | 294 ± 30.0 | 28.4 ± 9.9 | 8.4 ± 6.6 |

**Notes.**

ERP, event-related potential; VAS, visual analogue scale; SL, shallow lance stimulation with white noise; CL, clinical lance stimulation with white noise; ES, transcutaneous electrical stimulation with white noise; IES, intraepidermal electrical stimulation with white noise.

$\eta^2{}_p = 0.4076$) (Table 2). *Post hoc* analyses showed that N2–P2 amplitudes for the SL (33.0 ± 8.7 µV) and CL (38.1 ± 11.5 µV) were significantly larger than that for the ES (22.6 ± 4.3 µV: $P = 0.0025$ for SL; $P = 0.0001$ for CL). There was no significant difference in N2–P2 amplitudes between the SL and the CL.

The Friedman test indicated that the VAS scores differed significantly among the four stimulation conditions ($P < 0.0001$) (Table 2). *Post hoc* analyses indicated that the VAS score for the SL (6.4 ± 4.7 points) was significantly smaller than that for the CL (17.9 ± 10.6 points: $P < 0.0001$) and larger than that for the ES (3.8 ± 3.7 points: $P = 0.0348$). The VAS score for the CL was significantly larger than those for the ES ($P < 0.0001$) and IES (8.4 ± 6.6 points: $P < 0.0001$). In addition, the VAS score for the IES was significantly larger than that for the ES ($P = 0.0002$). There was no significant difference in the VAS scores between the SL and the IES. There were no correlations between the VAS scores and N2 or P2 latencies, or between the VAS scores and N2–P2 amplitudes, at any stimulation condition.

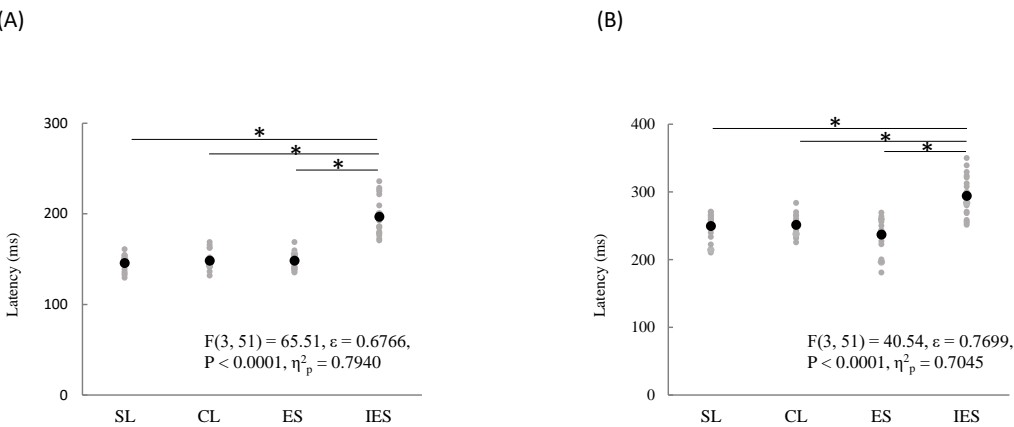

**Figure 5 N2 (A) and P2 latencies (B) for four stimulation condition (SL, CL, ES, and IES).** Black and gray circles indicate the mean and the individual data, respectively. SL, shallow lance stimulation with white noise; CL, clinical lance stimulation with white noise; ES, transcutaneous electrical stimulation with white noise; IES, intraepidermal electrical stimulation with white noise. Asterisks indicate significant *post hoc* differences.

## DISCUSSION

Here, we examined whether lance stimulation could selectively activate the free nerve endings of Aδ fibers. Specifically, we examined the effect of the click sound produced by the lance device on the ERPs evoked by skin puncture in Experiment 1, and compared the ERPs elicited by a custom-made shallow lance stimulation with those elicited by commonly clinically used lance stimulation, ES, and IES in Experiment 2. The present study revealed that the click sound shortened the latency of the ERP evoked by skin puncture and that the latencies of ERPs evoked by both lance stimulations (SL and CL) were shorter than those evoked by IES but similar to those evoked by ES.

### Effect of the click sound on the ERPs evoked by skin puncture

When a click sound was present (click-only and click+SL), the N2 latencies of ERPs in response to click-only and click+SL and the P2 latencies of ERPs in response to the click-only condition were within the range reported in previous studies investigating ERPs of auditory stimuli (*Wolpaw & Penry, 1975*; *Knight et al., 1980*; *Carrillo-de-la-Peña & Cadaveira, 2000*; *Atcherson et al., 2006*; *Gondan & Röder, 2006*; *Gondan, Vorberg & Greenlee, 2007*; *Touge et al., 2008*; *Sakamoto, Nakata & Kakigi, 2009*; *Sperdin et al., 2009*). N100, which can be observed in response to simple short auditory stimuli, reflects multiple mechanisms and is thought to originate from a wide range of areas (*Näätänen & Picton, 1987*; *Martin, Tremblay & Korczak, 2008*; *Sakamoto, Nakata & Kakigi, 2009*), such as Heschl's gyrus, the temporal lobe (*Okamoto et al., 2007*; *Altmann et al., 2008*), auditory association areas (*Knight et al., 1980*), and frontal areas (*Wolpaw & Penry, 1975*; *Atcherson et al., 2006*). P200, which can be observed following N100, has not been investigated to the same extent as N100, but is also thought to originate from a variety of areas in the cortex (*Wolpaw & Penry, 1975*; *Näätänen & Picton, 1987*; *Martin, Tremblay & Korczak, 2008*). Although the exact

mechanisms behind the ERPs evoked by auditory stimuli have not been fully determined, the click sound produced by the lance device appears to generate an auditory evoked potential.

## Possibility of multimodal ERPs evoked by lance stimulation

In the present study, the N2 latencies of ERPs generated by click+SL were similar to those of click-only, and the P2 latencies of ERPs in response to click+SL were similar to those of SL (without the click sound), indicating that lance stimulation elicits an ERP composed of multimodal stimuli, namely the click sound and skin puncture. In previous studies on multimodal brain responses, multimodal ERPs were reported to be not simply a sum of the ERPs of each unimodal stimulus (*Gondan & Röder, 2006*; *Gondan, Vorberg & Greenlee, 2007*; *Touge et al., 2008*; *Sperdin et al., 2009*; *Brett-Green et al., 2010*; *Dionne, Legon & Staines, 2013*). Specifically, in these studies, ERPs generated by unimodal stimuli (*i.e.*, auditory (A) or visual (V) alone) and ERPs of bimodal stimuli (*i.e.*, AV) were recorded, and the ERPs of bimodal stimuli were compared with the sum of the two unimodal ERPs (*i.e.*, A + V) (*Foxe et al., 2000*; *Gondan & Röder, 2006*; *Gondan, Vorberg & Greenlee, 2007*; *Touge et al., 2008*; *Brett-Green et al., 2010*). Their results revealed that the ERPs of bimodal stimuli did not match the sum of the two unimodal ERPs (*i.e.*, AV $\neq$ (A + V)) (*Foxe et al., 2000*; *Gondan & Röder, 2006*; *Gondan, Vorberg & Greenlee, 2007*; *Touge et al., 2008*; *Brett-Green et al., 2010*), meaning that the ERPs of bimodal stimuli reflect, in addition to the processing associated with the unimodal stimuli, brain activity related to the interaction of two different unimodal stimuli (*Gondan & Röder, 2006*; *Gondan, Vorberg & Greenlee, 2007*).

Similarly, previous studies on brain responses to multimodal auditory-somatosensory stimuli have shown that the N2 of ERPs of multimodal auditory-somatosensory stimuli were similar to those of unimodal auditory stimuli (click sound), while the P2 of ERPs of multimodal auditory-somatosensory stimuli were similar to those of unimodal somatosensory stimuli (electrical stimulation) (*Touge et al., 2008*; *Brett-Green et al., 2010*). Our finding is in agreement with this result, suggesting the interaction of auditory and somatosensory stimuli (*Gobbelé et al., 2003*). As the posterior parietal (*Gobbelé et al., 2003*), auditory (*Foxe et al., 2000*; *Foxe et al., 2002*; *Gobbelé et al., 2003*), and secondary somatosensory (SII) cortices (*Lam et al., 1999*; *Lütkenhöner et al., 2002*; *Gobbelé et al., 2003*) are reported to be involved in auditory-somatosensory information processing, these brain areas may have been activated during the lance stimulation. It can be speculated that lance stimulation (with the click sound) elicits a multimodal ERP of auditory and puncture stimuli, which is different from the sum of two unimodal ERPs and is considered to possibly indicate an interaction of these two stimuli.

## Activation of Aβ fibers by SL

The latencies of the ERPs of the ES and IES in this study were within the range reported in previous studies (*Inui et al., 2002*; *Mouraux, Iannetti & Plaghki, 2010*; *Otsuru et al., 2010*; *Kodaira, Inui & Kakigi, 2014*; *Mouraux, Marot & Legrain, 2014*; *Omori et al., 2017*; *Kirimoto et al., 2018*). The IES produces an electrical current in the epidermis and selectively

activates free nerve endings of Aδ fibers when low stimulation current intensities are applied (*Mouraux, Iannetti & Plaghki, 2010*). In contrast, the electrical current produced by the ES reaches the deeper dermis where receptors of Aβ fibers, which have a lower threshold than Aδ fibers, are located (*Mouraux, Iannetti & Plaghki, 2010*). These receptors respond to vibration, indentation, and forces applied to the skin (*Johnson, 2001*; *Macefield, 2005*). Therefore, the IES and ES used in the present study can be considered to have activated the free nerve endings of Aδ fibers in the epidermis and the receptors of Aβ fibers in the deeper dermis, respectively.

On the other hand, the latencies of the ERPs of the SL in the present study were not within a range reflecting the activation of Aδ fibers, even though the blade of the lance device was inserted to a depth of 0.1 mm into the epidermis where the free nerve endings of Aδ fibers are located, and the effect of click sound was removed. It is likely that the SL generated a penetration force when the blade was inserted into the skin, activating the Aβ fibers. The penetration force has been shown to be the largest force in the process of insertion (*Meyer et al., 2014*; *Leonardi, Viganò & Nicolucci, 2019*). For example, a venipuncture needle (27 and 30 gauge (G); 8 mm length) produces a penetration force of 0.23–0.34 N (*Egekvist & Arendt-Nielsen, 1999*), a hypodermic needle pen (31–34 G; 3.5–8.0 mm length) used for insulin injection produces a penetration force of 0.36–0.73 N (*Leonardi, Viganò & Nicolucci, 2019*), and a needle used for sclerotomy (23 G; 2.30–3.96 mm length) produces a penetration force of 0.48–2.16 N (*Meyer et al., 2014*). Although the blade of the SL has a different shape (about 0.1 mm in length and about 0.25 mm wide) from those needles listed above, we speculate that it generates a penetration force. Indeed, the N2 and P2 latencies of the lance ERPs in this study were similar to those of ERPs evoked by lance stimuli in previous studies (N2: $130 \pm 40$ ms, P2: $258 \pm 61$ ms (*Fabrizi et al., 2013*); N2: 102 ms, P2: 249.5 ms (*Fabrizi et al., 2016*)), and they were also within the range reported in previous studies examining ERPs generated by the stimulation of Aβ fibers (*Inui et al., 2002*; *Mouraux, Iannetti & Plaghki, 2010*; *Otsuru et al., 2010*). As reported in an early study comparing mechanical and transcutaneous ES, in short-latency somatosensory evoked potential components, such as N20, mechanical stimulation with a needle (pain) does not differ significantly from mechanical stimulation with a plastic ball (tactile) and transcutaneous ES (*Kakigi & Shibasaki, 1984*), supporting our argument. Meanwhile, one may argue that the increased N2–P2 amplitude in the CL and SL as compared to the ES can reflect the activation of Aδ fibers. However, given the latencies of N2 and P2, it is reasonable to assume that the CL and SL activated more cutaneous mechanoreceptors (*e.g.*, vibration receptors in the deeper dermis layer) than the ES. Therefore, it is most likely that the SL activated the Aβ fibers by the penetration force associated with blade insertion, even though the blade was inserted only into the epidermis.

## Mismatch between ERPs and pain perception

The pain score on the VAS for the IES was not different from that for the SL, but was smaller than that for the CL. Some previous studies have reported that ERP amplitude

is associated with the intensity of pain perception (*Beydoun et al., 1993*; *García-Larrea et al., 1997*; *Ohara et al., 2004*; *Iannetti et al., 2005*). However, nociceptive brain responses have been observed without pain perception in other studies in which ERP was evoked by the activation of Aδ fibers using IES that did not reach the pain threshold (*Hagiwara et al., 2018*), by laser stimulation under anesthesia in monkeys (*Baumgärtner et al., 2006*), and by unperceived painful laser stimulation in humans (*Lee, Mouraux & Iannetti, 2009*). Additionally, ERPs evoked by electric and laser stimulations are known to be modulated by stimuli expectations (*Hird et al., 2018*). Indeed, in the present study, we found no correlation between the VAS score and ERP latency or between the VAS score and ERP amplitude under any stimulation conditions. Thus, it seems to be difficult to evaluate pain using ERP amplitude.

Moreover, it has been reported that painful stimuli do not induce nociceptive brain responses when Aβ fibers are activated simultaneously (*Torquati et al., 2003*; *Rustamov et al., 2016*). In this case, the ES current reaches the dermis where Aβ fiber receptors are found with lower activation thresholds than those of the free endings of Aδ fibers. Hence, even if the subjects felt pain with the SL, the Aβ fiber receptors could have been activated initially, which may have masked the activation of Aδ fibers.

## CONCLUSIONS

The click sound produced by the lance device influences the ERPs evoked by skin puncture. Furthermore, the latency of the ERP of lance stimulation was shorter than that of IES and similar to that of ES, which suggests that Aβ fibers are activated by lance stimulation. Lance ERPs, therefore, may reflect the activation of Aβ fibers rather than Aδ fibers. A pain index that correctly and reliably reflects nociceptive processing must be developed to improve pain assessment and management in neonates.

## ACKNOWLEDGEMENTS

We would like to thank all the participants for their time and effort.

### Funding
This work was supported in part by the Japan Society for the Promotion of Science (JSPS) KAKENHI (grant numbers JP17K19818 and JP20K19132). The funders had no role in study design, data collection and analysis, decision to publish, or preparation of the manuscript.

### Grant Disclosures
The following grant information was disclosed by the authors:
Japan Society for the Promotion of Science (JSPS) KAKENHI: JP17K19818, JP20K19132.

### Competing Interests
The authors declare there are no competing interests.

## Author Contributions

- Yui Shiroshita conceived and designed the experiments, performed the experiments, analyzed the data, prepared figures and/or tables, authored or reviewed drafts of the paper, and approved the final draft.
- Hikari Kirimoto and Tatsunori Watanabe conceived and designed the experiments, authored or reviewed drafts of the paper, and approved the final draft.
- Keisuke Yunoki performed the experiments, analyzed the data, prepared figures and/or tables, and approved the final draft.
- Ikuko Sobue analyzed the data, authored or reviewed drafts of the paper, and approved the final draft.

## Human Ethics

The following information was supplied relating to ethical approvals (i.e., approving body and any reference numbers):

The study was approved by the Ethics Committee of Hiroshima University (approval number: E-2044).

## Data Availability

The raw measurements are available in the Supplemental Files.

## Supplemental Information

Supplemental information for this article can be found online at http://dx.doi.org/10.7717/peerj.12250#supplemental-information.

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
