# Peer review of "Event-related potentials evoked by skin puncture reflect activation of Aβ fibers: comparison with intraepidermal and transcutaneous electrical stimulations"

_PeerJ, doi:10.7717/peerj.12250_

## Round 0.1 · original submission · Major Revisions

Your manuscript has now been seen by three reviewers. You will see from their comments below that while they find your work of interest, some major points are raised.

We are interested in the possibility of publishing your study, but would like to consider your response to these concerns in the form of a revised manuscript before we make a final decision on publication.

We, therefore, invite you to revise and resubmit your manuscript, taking into account the points raised. Please highlight all changes in the manuscript text file.

Reviewer 1 ·

Basic reporting

no comment

Experimental design

This is an interesting and well-designed research. The findings in this study help further understand the pain induced by skin puncture.

Validity of the findings

An interindividual variability as well as an intraindividual variability seems to be very high in VAS in SL. I am wondering whether each trial of SL was appropriately performed. Are there the possibility that different trials in SL activated cutaneous sensory fibers differently? The authors should discuss a high inter/intraindividual variability in VAS in SL. I think the authors should show the correlation between VAS and the latency of ERP in each trial in each stimulation.

I believe upward is negative in Figure 2. Please describe this.

Additional comments

This is a study investigating ERPs evoked by skin puncture, intra-epidermal stimulation (IES) and transcutaneous electrical stimulation (ES). The authors found that the latency of ERP to lance stimulation was shorter than that to ES and much shorter than that to IES. They concluded that the lance ERP did not reflect pain processing.

This is an interesting and well-designed research. The findings in this study help further understand the pain induced by skin puncture. I have several comments that may strengthen this manuscript.

An interindividual variability as well as an intraindividual variability seems to be very high in VAS in SL. I am wondering whether each trial of SL was appropriately performed. Are there the possibility that different trials in SL activated cutaneous sensory fibers differently? The authors should discuss a high inter/intraindividual variability in VAS in SL. I think the authors should show the correlation between VAS and the latency of ERP in each trial in each stimulation.

I believe upward is negative in Figure 2. Please describe this.

Reviewer 2 ·

Basic reporting

no comment

Experimental design

no comment

Validity of the findings

no comment

Additional comments

The study was designed well and the conclusion drawn by the present results appears convincing. I have only a minor comment that I want the authors to consider.

Comment
In an earlier study, Kakigi and Shibasaki compared short-latency SEP components, such as N20, between mechanical and electrical stimulation of the middle finger and found that waveforms by mechanical stimulation with a needle did not differ significantly from those by mechanical stimulation with a ball or by electrical stimulation (Electroenceph clin Neurophysiol 1984). Authors would find it worth discussing in their paper.

·

Basic reporting

The manuscript is well structured and written clearly but should be checked for some small grammar errors such as missing pronouns. The Introduction provides the necessary background. Figures are relevant and correctly described.

The statement (abstract, but also introduction) that intra-epidermal stimulation (IES) “selectively activates A-delta fibers”, and that it is “well-established that pain processing can be evaluated using evoked potentials following the IES” should be rephrased more cautiously. First, the authors should take into consideration publications that have questioned the selectivity of IES (de Tommaso et al., 2011; La Cesa et al., 2017), and stress that IES is selective if and only if low stimulation currents are used. Second, I disagree with the statement that it is “well established” that IES can evaluate “pain processing”, and propose to replllace the term “pain processing” by “nociceptive processing” here and elsewhere in the manuscript (e.g., line 98: “ERPs reflecting pain (A-delta and C fibers)”).

The site of stimulation (hand dorsum) should be mentioned in the abstract as peripheral conduction distance is essential to interpret the reported ERP latencies. This also applies to the Introduction when discussing ERP latencies measured in adults (lines 79-99). Similarly, when comparing the latencies elicited by electric, mechanical and thermal stimuli, the authors should consider the impact of mechanical and heat transduction times.

Lines 77-78 : “whether heel lance in neonates activates A-delta or C fibers are unknown to date”. Consider rephrasing as it seems to me that what is unknown is whether the ERPs elicited by the heel lance reflect the activation of A-delta and/or C fibers, and not whether the heel lance stimulus activates these fibers.

Experimental design

Lance stimulation. Whether or not the release of the blade produces a “clicking” sound should be mentioned (see main comment in the next section).

It is mentioned that ES was delivered at “1.1-1.5 times the sensory threshold” and that IES was “1.5-2.0 times the sensory threshold”. It thus seems that the stimulation intensity was not determined solely based on sensory threshold. The authors should explain how the intensity of stimulation was determined for each participant.

Whether the experimenters were able to accurately monitor the onset of the lancet stimulus is crucial to interpret ERP latencies. Lines 185-189, it is mentioned that “an accelerometer attached to the experimenter’s finger detected the start of pressing the lancet”. Did the accelerometer detect the vibrations produced by the release of the blade? It is mentioned that “the duration from the lancet button press to the blade release was negligible as confirmed by a high-speed camera”. It would be worthwhile to report the actual delay that was measured by the camera recordings.

Validity of the findings

The results show clearly that the latencies of the responses elicited by the lance stimulus are shorter than the responses elicited by IES, supporting the author’s conclusion that lance-evoked responses probably do not reflect activity triggered by inputs conveyed by nociceptive A-delta fibers.

More surprising is the observation that the latencies of the responses elicited by the lance are also significantly shorter than the responses elicited by ES. What is the explanation for this difference if, as concluded by the authors, both lance stimulation and ES reflect activity conveyed by non-nociceptive A-beta fibers?

This brings me to my main concern which must be clarified by the authors. During lance stimulation, does the release of the blade produce a click sound? If so, could it be that the ERPs elicited by the lance stimulus reflect an auditory-evoked response rather than a somatosensory-evoked response? This would explain the early latency of the responses elicited by both SL and CL.

The discussion section on the “mismatch between ERP and pain perception” is not so relevant and very speculative. The authors may consider removing this section from the manuscript. Similarly, the discussion on analysis of “brain network activity” (Directions of future research) is not related directly to the present study.

Line 296, it is concluded that “an alternative method to evaluate the pain of skin puncture should be investigated”. The sentence could be rephrased because it is my impression that the main aim of recording heel lance evoked responses has been to investigate nociceptive processing in newborn infants rather than to “evaluate the pain of skin puncture”.

Additional comments

No additional comments.

---

## Round 0.2 · Minor Revisions

Your manuscript has now been seen by the reviewers. You will see from the comments below that an important point needs to be considered. We therefore invite you to revise and resubmit your manuscript, taking into account this point. Please highlight the changes in the manuscript text file.

Reviewer 1 ·

Basic reporting

no comment

Experimental design

no comment

Validity of the findings

no comment

Additional comments

I would like to thank the authors for addressing my initial comments.

They performed new experiments and showed new findings in the revised manuscript. I think the revised manuscript have problem as indicated below.

I disagree with their conclusion, “Lance ERPs, therefore, may not reflect nociceptive processing.” First, they should specify what they mean by “nociceptive processing”. Is it the activation of Aδ fibers, pain perception, or something else? Second, they found SL and CL, which activate Aδ fibers, had significantly larger N2-P2 amplitudes than ES which does not activate Aδ fibers. I think the difference may reflect “nociceptive processing”. They should discuss what caused the difference of the N2-P2 amplitudes.

---

## Round 0.3 · accepted · Accept

Thank you for the detailed response letter. We are delighted to accept your manuscript for publication.

Reviewer 1 ·

Basic reporting

no comment

Experimental design

no comment

Validity of the findings

no comment

Additional comments

Thank you for addressing my comments.